# Compressing Transformer-based Sequence to sequence Models with Pre-Trained Autoencoders for Text Summarization

## Abstract

We proposed a technique to reduce the decoder's number of parameters in a sequence-to-sequence (seq2seq) architecture for automatic text summarization. This approach uses a pre-trained autoencoder (AE) trained on top of a pre-trained encoder's output to reduce its embedding dimension and allow to significantly reduce the summarizer model's decoder size. The ROUGE score is used to measure the effectiveness of this method by comparing four different latent space dimensionality reductions: 96%, 66%, 50%, 44%. A few well-known frozen pre-trained encoders (BART, BERT, and DistilBERT) have been tested, paired with the respective frozen pre-trained AEs to test the reduced dimension latent space's ability to train a summarizer model. We also repeated the same experiments on a small transformer model that has been trained for text summarization. This study shows an increase of the R-1 score by 5% while reducing the model size by 44% using the DistilBERT encoder, and competitive scores for all the other models associated to important size reduction. It is also shown that our approach can be used in combination with other network size reduction techniques (e.g. Distillation) to further reduce any encoder-decoder model parameters count.

## 1 Introduction

It is safe to say that the combination of Transformer (Vaswani et al., 2017) architecture, and transfer learning concept dramatically modified the landscape of Natural Language Processing (NLP). Introduction of large-scale pre-trained language models like BERT (Devlin et al., 2018), GPT-2 (Radford et al., 2019), MegatronLM (Shoeybi et al., 2019), BART (Lewis et al., 2019), and GPT-3 (Brown et al., 2020) keeps on improving state-of-the-art results by fine-tuning them for downstream tasks such as Sentiment Analysis, Question Answering, and Summarization. However, the upward trend of the network size in mentioned models raises serious environmental (Strubell et al., 2019) and usability issues.

The number of parameters in Transformers-based models is constantly rising. As a result, the hardware requirement for fine-tuning or inference drastically increased in the past couple of years. It is challenging for both researchers and developers to use these models and build on top of them without sufficient resources. To put it in perspective, the earliest and smallest BERT had 110M parameters (Devlin et al., 2018), and the latest and largest Switch Transformer (Fedus et al., 2021) model was introduced with 1.5 trillion parameters which makes it accessible only using high-end servers. This issue is even more consequential for tasks such as automatic text summarization and machine translation that incorporate sequence-to-sequence architecture. This architecture consists of an encoder (encoding the input sequences) paired with a decoder (generates tokens conditioned on the encoder's representation). It means even more parameters will be added to the model for these tasks which is the min focus of this paper.

It is important to discover techniques to reduce the overall network parameters while maintaining the quality of the generated text. In this paper, autoencoders' (AE) (Liu et al., 2019) property of dimensionality reduction is evaluated in a setting with sequence-to-sequence architecture and pre-trained encoders. The autoencoder will act as an intermediate model to compress the encoder's final representation and decoder will use this compressed latent representation to generate summaries

with minimal information loss. The idea is to find the ideal trade-off between the compression ratio and model's text generation capability.

## 2 BACKGROUND

Multiple different approaches have already been explored to tackle the problem of neural network models growing size. Quantization (Gupta et al., 2015) is one of the first experiments which apply to any deep learning model by using a half-precision (16-bit) floating point to greatly reduce the network size and memory usage. Micikevicius et al. proposed a mixed precision algorithm in (Micikevicius et al., 2017) to further close the gap in evaluation results. Recent works experimented the effect of knowledge distillation (Bucila et al., 2006) method to transfer information from a larger network to a smaller one without significant loss in accuracy. There are multiple papers that present different combinations of fine-tuning and distillation on top of BERT. (Chatterjee, 2019; Turc et al., 2019) However, DistilBERT (Sanh et al., 2019) obtained the best results with training the smaller student model on BERT and then fine-tuning it for downstream tasks that resulted in a more generalized pre-trained model. Their approach led to similar implementations on other classic transformer models such as DistilGPT2 with 33% less parameters (two times faster) which resulted in 21.1 perplexity score comparing to the GPT-2's 16.3, and a 35% smaller RoBERTa (Liu et al., 2019) model while maintaining 95% of the accuracy on GLUE, named DistilRoBERTa.[1]

The pruning (LeCun et al., 1990) method's influence on transfer learning have recently gained attention from researchers. It refers to determining the parts in the network that have the weaker effect on the model accuracy and removing them without compromising the model's accuracy on downstream tasks. The main ideas are to either focus on finding the less important weights (Gordon et al., 2020), or components such as number of self-attention heads (Michel et al., 2019) and layers (Sajjad et al., 2020). This technique was also used in the Lottery Ticket hypothesis (Frankle & Carbin, 2018; Prasanna et al., 2020) to uncover subnetworks performing on par with the full model.

The latest research area focuses on rethinking the self-attention mechanism to eliminate its quadratic memory usage connection with respect to the input sequence length. The goal is to find the best trade-off between performance and memory usage. Big Bird (Zaheer et al., 2020) and Longformer (Beltagy et al., 2020) papers experiment on different attention patterns to reduce connections and result in fewer computations. Wang et al. presented the Linformer (Wang et al., 2020) network that projects the self-attention vectors to lower dimensions. Reformer (Kitaev et al., 2020) paper studies the idea of grouping key and query vectors based on the locality sensitive hashing (LSH) to reduce the computations needed to find similar vectors.

It is worth noting several studies that combined two or more methods to build even smaller models without significantly compromising the accuracy. Several such experiments are present in the literature, namely, a combination of distillation with pruning (Hou et al., 2020), or quantization (Sun et al., 2020). Also, Tabmbe et al. made the EdgeBERT (Tambe et al., 2020) model by leveraging both pruning and quantization along with other methods. These techniques are not exclusive, and it is possible to study them independently without testing all possible combinations. This is why we can focus our study only on one reduction method and only a few important models.

## 3 PROPOSED METHOD

Our proposed architecture is a sequence-to-sequence Transformer ($T$) model that includes a pre-trained autoencoder (AE) connecting the network's encoder ($T_{enc}$) to its decoder ($T_{dec}$). (Fig. 1) The mentioned approach will result in a smaller $T_{dec}$ and reduce the overall number of trainable parameters. The modules are described in the sub-sections.

### 3.1 TRANSFORMER ENCODER ($T_{enc}$)

The encoder ($T_{enc}$) part of the proposed architecture is a pre-trained transformer-based model. We have selected a set of models that include some of the best models for text summarization; BERT,

---

[1]The results related to DistilRoBERTa model is available at `https://github.com/huggingface/transformers/tree/master/examples/research_projects/distillation`

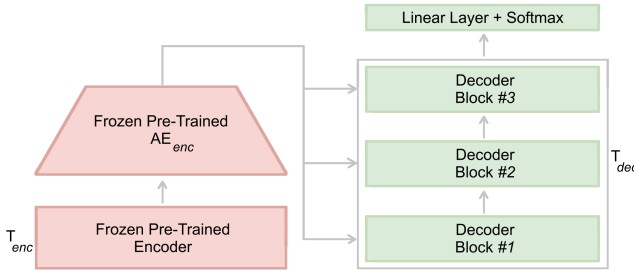

Figure 1: The proposed architecture. The red-colored components in the diagram indicates being both pre-trained, and frozen during training the summarizer model. The green-colored units are learned from scratch for summarization task.

DistilBERT, BART's encoder (all base versions), and a custom transformer model. The custom transformer model is used to evaluate the effectiveness of our approach on a small pre-trained model with only 6 encoder layers and subsequently its scores are used as baseline and not supposed to be competitive with the ones of the other approaches. These models are frozen during the training processes to reduce the number of influential factors.

## 3.2 AUTOENCODER (AE)

The AE (Fig. 1) purpose is to reduce the $T_{enc}$'s output size to a smaller latent space using the following equation:

$$X'_{AE} = AE_{dec}(AE_{enc}(X_{AE}))$$
$$Z = AE_{enc}(X_{AE})$$

Where $X_{AE}^{S \times D}$ is the input, $AE_{enc}$ indicates the encoder responsible for compressing the input to latent space $Z^{S \times C}$ and a decoder $AE_{dec}$ generating the output $X'^{S \times D}_{AE}$ trying to reconstruct the input $X_{AE}^{S \times D}$ during the training process. Variables $S$ and $D$ denote the sequence length and input's embedding dimension respectively. The values are dependent to the chosen pre-trained $T_{enc}$ model's configuration and are set to 512 and 768 in this paper. However, $C$ that represents the compressed latent space size, will vary to find out the optimal latent space size. A 6-layer linear AE (3 for encoder and 3 for decoder) was selected after comparing its reconstruction ability to the same architecture with Long Short-Term Memory (LSTM) (Hochreiter & Schmidhuber, 1997), or Convolutional Neural Network (CNN) (LeCun et al., 1998) building blocks.

The final AE architecture with 6 linear layers was independently trained for each selected pre-trained encoder $T_{enc}$. It attempts to reconstruct the output of $T_{enc}$ using a smaller representation $Z$. The frozen $AE_{enc}$ is then used in our summarizer architectures to pass a compressed representation to the decoder $T_{dec}$. (From size $D$ to $C$) Refer to appendix A.1 for more information about the hidden layers' sizes.

## 3.3 TRANSFORMER DECODER ($T_{dec}$)

The decoder component of the architecture ($T_{enc}$) is an original transformer decoder with 3-layers and a linear head on top in all the experiments. It is the only piece of the network that is not frozen after the AE has been trained. Its embedding dimension ties to the AE's latent space size ($C$) that can drastically alter the architecture overall number of trainable parameters.

## 3.4 DATASETS

A combination of the CNN/Dailymail (300K samples) (Hermann et al., 2015) and the Newsroom (1.3M samples) (Grusky et al., 2018) datasets used for training all the summarizer models; and the

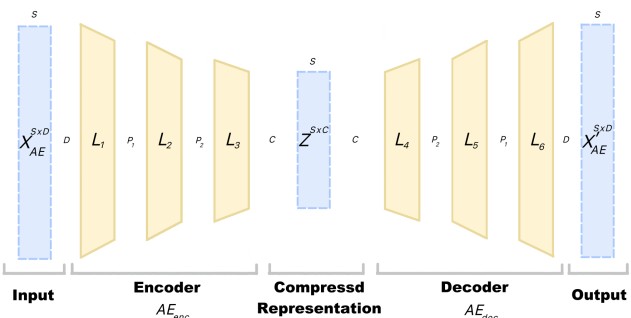

Figure 2: The linear autoencoder architecture with 3 encoder (L1, L2, L3), and decoder (L4, L5, L6) layers. Tensors $X$, $X'$, and $Z$ are representing the input, output, and the compressed latent representation respectively. The autoencoder maintain the same sequence length ($S$) during compression and only reduce the embedding size ($D$).

pre-defined test set of CNN/Dailymail dataset utilized to evaluate them. However, we randomly selected only 60% of these combinations to train each autoencoder models to hold on unseen data to evaluate the generalization ability of the summarizer model.

## 3.5 EXPERIMENTS

We performed several experiments to evaluate the effectiveness of the proposed method. First (AE), we combined different pre-trained encoders with several autoencoder compressed latent space sizes paired with a 3-layers decoder component in each instance. Second (AE-S), we used the same autoencoders (without the pre-training step) and trained them jointly with the decoder from scratch to measure the effect of the pre-training step on the autoencoder. Third (LL), using a small 1-layer learnable linear model to lower the encoder's output dimensionality from $D$ to $C$. Lastly (PCA), we used the classical dimensionality reduction algorithm, incremental PCA (Ross et al., 2008) trained to project the outputs of the encoder to the 458 first principal components to preserve more than 90% of variances and used them as the decoder input.

All these models are trained with fixed hyperparameters to make them comparable. They are trained using the 1cycle (Smith & Topin, 2019) training policy that increases the learning rate from 2e-5 to a maximum of 5e-4 while decreasing the momentum beta from 0.95 to 0.85 for faster convergence. We also used the Adam optimizer and the label smoothing CrossEntropy (Pereyra et al., 2017) loss function. The decoder's feedforward layers and the eight attention heads' inner dimensions are 1024 and 32, respectively, and their input sizes are dependent on the dimensionality reduction rate.

## 4 RESULTS

The summaries are generated using three different decoding strategies. The fastest and easiest method to develop is picking the most probable output at each timestep, known as the Greedy algorithm. A more extensive approach is to use the Beam Search algorithm that develops K paths in parallel using the top K tokens with the highest scores. Finally, we used the Weighted Random Sampling algorithm that randomly chose a token from the top K probable outputs proportionally to their respective probabilities. In this paper, we used K value equals 5 and 2 for the beam search and weighted random sampling, respectively, based on the previous experiments.

Table 1: The comparison of MSE loss between linear, LSTM, and CNN blocks to train an autoencoder with a 64 compression size.

| Types | Number of Layers | 4 | 6 | 8 |
|---|---|---|---|---|
| Linear | | 0.0813 | 0.0766 | 0.0776 |
| LSTM | | 0.0863 | 0.0810 | 0.0849 |
| CNN | | 0.2666 | 0.2759 | 0.2750 |

We experimented with different AE layer types and numbers. Linear, Long Short-Term Memory (LSTM), and Convolutional Neural Network (CNN) AE architectures were trained with the Mean

Table 2: Comparing the number of parameters in a 3-layer decoder network with 768 input size to the number of parameters of the same decoder after applying the AutoEncoder to reduce the encoder's output dimension.

| Decoder Input Size | AE Parameters Count | Decoder Parameters Count (by Encoder Type) | | | | Network's Total Number of Parameters (by Encoder Type) | | | |
|---|---|---|---|---|---|---|---|---|---|
| | | BERT/ Transformer | Reduction (%) | BART | Reduction (%) | Transformer | BART | BERT | DistilBERT |
| $C = 768$ (**Default**) | - | 32,937,018 | - | 48,119,385 | - | 70,572,090 | 188,119,385 | 142,937,018 | 98,937,018 |
| $C = 512$ | 2,316,288 | 21,970,746 | 26.26 | 32,098,905 | 28.48 | 61,922,106 | 174,415,193 | 134,287,034 | 90,649,914 |
| $C = 384$ | 1,812,480 | 16,487,610 | 44.44 | 24,088,665 | 46.18 | 55,935,162 | 165,901,145 | 128,300,090 | 84,662,970 |
| $C = 128$ | 1,120,256 | 5,521,338 | 79.84 | 8,068,185 | 80.91 | 44,276,666 | 149,188,441 | 116,641,594 | 75,484,564 |
| $C = 32$ | 1,071,104 | 1,408,986 | 92.47 | 2,060,505 | 93.5 | 40,115,162 | 143,131,609 | 112,480,090 | 68,842,970 |

Table 3: The ROUGE score for using a pre-trained autoencoder on top of pre-trained transformer-based encoders with different compression sizes. Tested each network using greedy, weighted random sampling, and beam search methods.

| Models | Inference Methods | | | | | | | | | | | | | | |
|---|---|---|---|---|---|---|---|---|---|---|---|---|---|---|---|
| | Greedy | | | | | Random | | | | | Beam | | | | |
| | R-1 | R-2 | R-3 | R-L | R-W | R-1 | R-2 | R-3 | R-L | R-W | R-1 | R-2 | R-3 | R-W | R-L |
| **Transformer** | 0.346 | 0.143 | 0.079 | 0.312 | 0.121 | 0.344 | 0.136 | 0.071 | 0.304 | 0.117 | 0.259 | 0.116 | 0.064 | 0.261 | 0.095 |
| **+ AE** | 0.368 | 0.157 | 0.088 | 0.325 | 0.129 | 0.363 | 0.147 | 0.079 | 0.315 | 0.123 | 0.288 | 0.127 | 0.070 | 0.280 | 0.104 |
| ($C = 512$) | (106%) | (109%) | (111%) | (104%) | (106%) | (105%) | (108%) | (111%) | (103%) | (105%) | (111%) | (109%) | (109%) | (107%) | (109%) |
| **+ AE** | 0.363 | 0.154 | 0.086 | 0.322 | 0.127 | 0.360 | 0.146 | 0.078 | 0.314 | 0.122 | 0.278 | 0.123 | 0.068 | 0.274 | 0.101 |
| ($C = 384$) | (104%) | (107%) | (108%) | (103%) | (104%) | (104%) | (107%) | (109%) | (103%) | (104%) | (107%) | (106%) | (106%) | (104%) | (106%) |
| **+ AE** | 0.308 | 0.114 | 0.060 | 0.286 | 0.109 | 0.315 | 0.110 | 0.054 | 0.280 | 0.106 | 0.280 | 0.116 | 0.064 | 0.271 | 0.100 |
| ($C = 128$) | (89%) | (79%) | (75%) | (91%) | (90%) | (91%) | (80%) | (76%) | (92%) | (90%) | (108%) | (100%) | (100%) | (103%) | (105%) |
| **+ AE** | 0.156 | 0.019 | 0.003 | 0.174 | 0.057 | 0.184 | 0.024 | 0.003 | 0.185 | 0.061 | 0.132 | 0.021 | 0.004 | 0.147 | 0.045 |
| ($C = 32$) | (45%) | (13%) | (3%) | (55%) | (47%) | (53%) | (17%) | (4%) | (60%) | (52%) | (50%) | (18%) | (6%) | (56%) | (47%) |
| **BART** | 0.355 | 0.142 | 0.076 | 0.310 | 0.121 | 0.349 | 0.134 | 0.069 | 0.301 | 0.116 | 0.304 | 0.128 | 0.070 | 0.283 | 0.106 |
| **+ AE** | 0.341 | 0.128 | 0.066 | 0.298 | 0.114 | 0.337 | 0.120 | 0.058 | 0.289 | 0.110 | 0.312 | 0.126 | 0.068 | 0.285 | 0.107 |
| ($C = 512$) | (96%) | (90%) | (86%) | (96%) | (94%) | (96%) | (89%) | (84%) | (96%) | (94%) | (102%) | (98%) | (97%) | (101%) | (101%) |
| **+ AE** | 0.332 | 0.121 | 0.061 | 0.291 | 0.111 | 0.327 | 0.112 | 0.053 | 0.282 | 0.106 | 0.278 | 0.123 | 0.068 | 0.274 | 0.101 |
| ($C = 384$) | (93%) | (85%) | (80%) | (93%) | (91%) | (93%) | (83%) | (76%) | (93%) | (91%) | (91%) | (96%) | (97%) | (96%) | (95%) |
| **+ AE** | 0.257 | 0.063 | 0.023 | 0.239 | 0.084 | 0.260 | 0.058 | 0.019 | 0.232 | 0.081 | 0.245 | 0.071 | 0.030 | 0.234 | 0.080 |
| ($C = 128$) | (72%) | (44%) | (30%) | (77%) | (69%) | (74%) | (43%) | (27%) | (77%) | (69%) | (80%) | (55%) | (42%) | (82%) | (75%) |
| **+ AE** | 0.145 | 0.014 | 0.001 | 0.168 | 0.055 | 0.174 | 0.019 | 0.002 | 0.179 | 0.059 | 0.128 | 0.015 | 0.002 | 0.146 | 0.045 |
| ($C = 32$) | (40%) | (9%) | (1%) | (54%) | (45%) | (49%) | (14%) | (2%) | (59%) | (50%) | (42%) | (11%) | (2%) | (51%) | (42%) |
| **BERT** | 0.349 | 0.133 | 0.066 | 0.306 | 0.117 | 0.347 | 0.124 | 0.058 | 0.297 | 0.112 | 0.283 | 0.117 | 0.060 | 0.270 | 0.099 |
| **+ AE** | 0.339 | 0.123 | 0.058 | 0.298 | 0.114 | 0.339 | 0.116 | 0.051 | 0.289 | 0.108 | 0.291 | 0.117 | 0.059 | 0.275 | 0.100 |
| ($C = 512$) | (97%) | (92%) | (87%) | (97%) | (97%) | (97%) | (93%) | (87%) | (97%) | (96%) | (102%) | (100%) | (98%) | (101%) | (101%) |
| **+ AE** | 0.332 | 0.119 | 0.056 | 0.294 | 0.110 | 0.333 | 0.112 | 0.048 | 0.286 | 0.106 | 0.272 | 0.107 | 0.053 | 0.263 | 0.094 |
| ($C = 384$) | (95%) | (89%) | (84%) | (96%) | (94%) | (95%) | (90%) | (82%) | (96%) | (94%) | (96%) | (91%) | (88%) | (97%) | (94%) |
| **+ AE** | 0.278 | 0.074 | 0.025 | 0.256 | 0.090 | 0.288 | 0.072 | 0.022 | 0.252 | 0.088 | 0.242 | 0.076 | 0.029 | 0.237 | 0.080 |
| ($C = 128$) | (79%) | (55%) | (37%) | (83%) | (76%) | (82%) | (58%) | (37%) | (84%) | (78%) | (85%) | (64%) | (48%) | (87%) | (80%) |
| **+ AE** | 0.168 | 0.021 | 0.003 | 0.187 | 0.062 | 0.197 | 0.026 | 0.003 | 0.194 | 0.064 | 0.140 | 0.020 | 0.003 | 0.153 | 0.047 |
| ($C = 32$) | (48%) | (15%) | (4%) | (61%) | (52%) | (56%) | (20%) | (5%) | (65%) | (57%) | (49%) | (17%) | (5%) | (56%) | (47%) |
| **DistilBERT** | 0.317 | 0.124 | 0.064 | 0.283 | 0.100 | 0.316 | 0.117 | 0.057 | 0.275 | 0.097 | 0.302 | 0.123 | 0.066 | 0.280 | 0.099 |
| **+ AE** | 0.333 | 0.123 | 0.059 | 0.298 | 0.112 | 0.332 | 0.116 | 0.052 | 0.290 | 0.108 | 0.287 | 0.116 | 0.059 | 0.275 | 0.100 |
| ($C = 512$) | (105%) | (99%) | (92%) | (105%) | (112%) | (105%) | (99%) | (91%) | (105%) | (111%) | (95%) | (94%) | (89%) | (98%) | (101%) |
| **+ AE** | 0.334 | 0.122 | 0.060 | 0.297 | 0.112 | 0.334 | 0.115 | 0.052 | 0.288 | 0.108 | 0.282 | 0.114 | 0.058 | 0.270 | 0.097 |
| ($C = 384$) | (105%) | (98%) | (93%) | (104%) | (112%) | (105%) | (98%) | (91%) | (104%) | (111%) | (93%) | (92%) | (87%) | (96%) | (97%) |
| **+ AE** | 0.287 | 0.083 | 0.031 | 0.265 | 0.094 | 0.297 | 0.081 | 0.028 | 0.259 | 0.092 | 0.240 | 0.082 | 0.035 | 0.238 | 0.081 |
| ($C = 128$) | (90%) | (66%) | (48%) | (93%) | (94%) | (93%) | (69%) | (49%) | (94%) | (94%) | (79%) | (66%) | (53%) | (85%) | (81%) |
| **+ AE** | 0.161 | 0.020 | 0.003 | 0.180 | 0.059 | 0.189 | 0.024 | 0.003 | 0.188 | 0.062 | 0.134 | 0.019 | 0.003 | 0.147 | 0.045 |
| ($C = 32$) | (50%) | (16%) | (4%) | (63%) | (59%) | (59%) | (20%) | (5%) | (68%) | (63%) | (44%) | (15%) | (4%) | (52%) | (45%) |

Square Error (MSE) loss for seven epochs to measure their performance. A latent space representation of size 64 was selected for this comparison benchmark (Table 1). The linear autoencoder outperforms both LSTM and CNN in all experiments. (Refer to Appendix A.1, Table 6 for the full list of all compression rates) Also, the 6-layer design choice results in a better score in all experiments.

The model size comparisons (number of parameters) are presented in Table 2. The first step is to calculate the decoder's size without using the dimensionality reduction method. As previously mentioned, based on the choice of the pre-trained encoder models, the default encoder's representation dimension is 768 which results in a decoder with either 48M (for BART encoder) or 33M (other options) parameters. We use this value as the reference number to measure the reduction percentage. It is important to keep in mind that even though there are parameters being added to the model from the autoencoder that affects the percentage, the number of trainable parameters in all the experiments is equal to the decoder size since the rest of the components (encoder, and autoencoder) are frozen during the tests. The reason why we have decoders with different parameter counts while using the same hyper-parameters is because we use each model's pre-trained tokenizers which have different vocabulary size, which affects the decoder's last layer output dimension. BERT, DistilBERT, and Transformer models use the BERT's pre-trained tokenizer, and BART uses its own.

Table 4: The comparison between using the pre-trained AutoEncoder (AE), training the AutoEncoder's encoder jointly with the network from scratch (AE S), using a simple linear layer model for the projection (LL), and PCA to do the dimensionality reduction.

| Models | Greedy | | | | | Random | | | | | Beam | | | | |
|---|---|---|---|---|---|---|---|---|---|---|---|---|---|---|---|
| | R-1 | R-2 | R-3 | R-L | R-W | R-1 | R-2 | R-3 | R-L | R-W | R-1 | R-2 | R-3 | R-W | R-L |
| **BERT** | 0.349 | 0.133 | 0.066 | 0.306 | 0.117 | 0.347 | 0.124 | 0.058 | 0.297 | 0.112 | 0.283 | 0.117 | 0.060 | 0.270 | 0.099 |
| **+ AE** | 0.339 | 0.123 | 0.058 | 0.298 | 0.114 | 0.339 | 0.116 | 0.051 | 0.289 | 0.108 | 0.291 | 0.117 | 0.059 | 0.275 | 0.100 |
| ($C = 512$) | (97%) | (92%) | (87%) | (97%) | (97%) | (97%) | (93%) | (87%) | (97%) | (96%) | (102%) | (100%) | (98%) | (101%) | (101%) |
| **+ AE S** | 0.289 | 0.079 | 0.027 | 0.262 | 0.093 | 0.300 | 0.079 | 0.025 | 0.258 | 0.092 | 0.26 | 0.084 | 0.033 | 0.250 | 0.086 |
| ($C = 512$) | (82%) | (59%) | (40%) | (85%) | (79%) | (86%) | (63%) | (43%) | (86%) | (82%) | (92%) | (71%) | (55%) | (92%) | (86%) |
| **+ LL** | 0.277 | 0.083 | 0.033 | 0.260 | 0.092 | 0.283 | 0.080 | 0.030 | 0.254 | 0.090 | 0.234 | 0.083 | 0.036 | 0.238 | 0.081 |
| ($C = 512$) | (79%) | (62%) | (50%) | (84%) | (78%) | (81%) | (64%) | (51%) | (85%) | (80%) | (82%) | (70%) | (60%) | (88%) | (81%) |
| **+ PCA** | 0.143 | 0.016 | 0.002 | 0.156 | 0.046 | 0.158 | 0.017 | 0.002 | 0.159 | 0.048 | 0.116 | 0.013 | 0.002 | 0.131 | 0.038 |
| ($C = 458$) | (40%) | (12%) | (3%) | (50%) | (39%) | (45%) | (13%) | (3%) | (53%) | (42%) | (40%) | (11%) | (3%) | (48%) | (38%) |

*Inference Methods*

Table 5: The comparison of a few vanilla encoder-decoder model generated summaries to the ones generated by the same model with the addition of AE with a latent space size of 384. Both results are generated using the greedy decoding method.

| Model | Vanilla Model Generated Summary (Greedy) | +AE Model Generated Summary (Greedy) |
|---|---|---|
| **Transformer** | masked men armed with handguns have robbed three banks in pittsburgh area . they are believed to have had military training and are being described as ' armed and extremely dangerous ' . the men are believed to have been threatened to kidnapping those at their targets and shoot police . however , the way that the men handle | two men armed with handguns robbed three banks in pittsburgh area so far this year . the unknown men , who are seen on surveillance footage pointing their guns at bank employees ' heads , have threatened to kidnapping those at their targets and shoot police . however , the way the men handle their weapons has led |
| **BART** | two men are believed to have had military training and are being described by the fbi as 'armed and extremely dangerous'. the men are seen holding their finger stretched along the barrel of his gun, just off of the trigger, a safety method used by law enforcement. the men, who wear dark | two pennsylvania bank robber are believed to have had military training. they are believed to have been armed and extremely dangerous. the men are believed to have been armed with a pair of masked men armed with handgun. the men are believed to have been from pittsburgh. |
| **BERT** | two pennsylvania bank robbers have robbed three banks in the pittsburgh area so far this year . the unknown men , who are seen on surveillance footage , have threatened to kidnapping those at their targets and shoot police . the two men , both 5 ' 5 ' - 9 ' and april 10 , are described as | two pennsylvania bank robbers armed with handguns have been robbed in the pittsburgh area so far this year . they have been seen jumping over the counter as they take their guns at targets and shoot them at police . the two men , who are seen on surveillance footage , have threatened to kidnapping those at their |
| **DistilBERT** | two robbers have been seen in a series of recent heisting robberies . the men are believed to have had military training and are being described as ' armed and extremely dangerous ' . the men are believed to have been armed and armed . the men are believed to have been armed and extremely dangerous . | two masked men armed with handguns have robbed three banks in the pittsburgh area so far this year . they are believed to have had military training and are being described by fbi as ' armed and extremely dangerous ' . the men are believed to have had military training and are being described by the fbi as |

The proposed method results on the text summarization task are shown in Table 3 where the classic ROUGE (R-1, R-2, R-3, R-L, and R-W) metric is reported using the greedy, weighted random sampling, and beam search decoding strategies. Autoencoders are trained with four (32, 128, 384, 512) different latent space dimensions ($C$) to analyze the effect of these dimensionality reductions. The models are also evaluated using the BERTScore (Zhang et al., 2019) metric and the results follow the same direction. (Refer to Appendix A.4)

Our configurations not only reduce the number of parameters in the network by reducing the decoder size, but also shows the ability to increase the model's summarization ability with a higher score compared to the original setup for the Transformer and DistilBERT models. The experiments show that adding an AE with $C = 512$ outperforms the same vanilla encoder-decoder network. Both BART and BERT experiments with the same AE size outperformed their vanilla model in several metrics only using the beam search method; however, the rest of the scores with a smaller decoder are still competitive. An even more surprising result is that a smaller dedicated Transformer with the proposed method and $C = 384$ performed better than BART and BERT in all the summarization benchmarks.

It also worth noting that the greedy inference method constantly results in better scores, with the weighted random sampling method following closely. The fact that beam search algorithm leans towards shorter sequences (Wu et al., 2016) reduces the ROUGE scores since there are fewer matching N-grams in the generated and target summaries. It does not mean that the sentences structure/quality are flawless using greedy/weighted random sampling, or poor using beam search, the results just reflect what the ROUGE score is measuring: an N-gram overlap between the generated and the target sequences.

Our ablation results show (Table 4) the impact of autoencoder pre-training step on the final score. It surpasses both training the $AE_{enc}$ jointly with the network from scratch (AE S) and using a simple

learnable linear layer for projecting (LL). Lastly, the PCA dimensionality reduction technique does not produce desirable results. The results also show that using an autoencoder with a latent space size of 384 ($C = 384$) generates ROUGE score close to the vanilla model. It reduces the decoder size by 46% from 48M to 24M for the BART model and 44% from 33M to 16M parameters for the other models. The critical point is that the combination of this configuration associated with greedy decoding algorithm shows no noticeable degrading quality in the generated summary. (Table 5)

## 5 CONCLUSION

In this paper, we presented a method to use pre-trained encoders models in a sequence-to-sequence setting while training a small decoder from a compress representation of the decoder's output for automatic text summarization task. The proposed architecture is based on an autoencoder pre-trained to reduce the encoder's representation size. The resulting compressed latent representation is used as inputs for a decoder. The main idea is that decreasing the size of the AE latent representation leads to dramatic decrease in the decoder size. We have shown that by reducing the encoder's output dimension from 768 to 384, not only we can reduce the decoder size by 44%, but it will also keep 95% of the R-1 score in the worst case (BERT) or increase it up to 105% with DistilBERT. Moreover, even with almost 80% reduction in the decoder size we can still keep 90% of the ROUGE score with our dedicated transformer and DistilBERT architectures.

Our method can be directly used together with other approaches such as distillation, pruning and quantization to further reduce the network size. One of our future research projects will be to investigate which of such combination could lead to the best compromise between the size and the accuracy of the model. It might also be interesting to study the effectiveness of our approach on other generative tasks like translation and question answering for further future works.

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

# A  APPENDIX

## A.1  AUTOENCODER ARCHITECTURE

Table 6 shows the full results of the different autoencoders building blocks and sizes. The linear layer AE outperforms both LSTM and CNN in almost all the compression sizes. While the only exception is the LSTM network with the smallest latent space dimension (32) that narrowly achieved better accuracy, none of the experiments had an acceptable performance in that representation size. Also, it shows that the CNN architecture resulted in the worst score by a large margin.

Table 7 shows the details of the 6-layer linear architecture that consists of 3 projections in each encoder and decoder. (Illustrated in Fig. 2) The idea is to keep a gentle decrease in size for large latent spaces, and to have enough learning capacity with wider networks in smaller compressed sizes.

Table 6: The MSE loss value of the selected 3 network types (LSTM, linear, CNN) with a different number of layers.

| Type | Compression Rate | Number of Layers | MSE Loss |
|---|---|---|---|
| | 32 | 6 | 0.0905 |
| | | 2 | 0.1176 |
| | | 4 | 0.0863 |
| | 64 | 6 | 0.0810 |
| | | 8 | 0.0849 |
| LSTM | | 10 | 0.1043 |
| | 128 | 6 | 0.0670 |
| | 256 | 6 | 0.0543 |
| | 384 | 6 | 0.0462 |
| | 448 | 6 | 0.0427 |
| | 512 | 6 | 0.0400 |
| | 32 | 6 | 0.0930 |
| | | 4 | 0.0752 |
| | 64 | 6 | 0.0766 |
| | | 8 | 0.0775 |
| Linear | 128 | 6 | 0.0637 |
| | 256 | 6 | 0.0453 |
| | 384 | 6 | 0.0278 |
| | 448 | 6 | 0.0239 |
| | 512 | 6 | 0.0200 |
| | | 4 | 0.2666 |
| CNN | 64 | 6 | 0.2759 |
| | | 8 | 0.2750 |

Table 7: The autoencoder models projections for different compression rates.

| First Projection (P1) | Second Projection (P2) | Third Projection / Compressed Latent Space Size (C) |
|---|---|---|
| 640 | 576 | 512 |
| 608 | 528 | 448 |
| 576 | 480 | 384 |
| 640 | 320 | 256 |
| 512 | 256 | 128 |
| 512 | 256 | 64 |
| 512 | 256 | 32 |

## A.2 Extra Results for BERT model + AE

We did a few more experiments on the BERT model to find the optimal latent space size and ROUGE score combination. As shown in Table 8, the intermediate $C$ values are also following the same trend as presented in Table 6.

Table 8: The ROUGE score of more experiments on the optimal latent space size options.

| Models | Inference Methods | | | | | | | | | | | | | | |
| --- | --- | --- | --- | --- | --- | --- | --- | --- | --- | --- | --- | --- | --- | --- | --- |
| | Greedy | | | | | Random | | | | | Beam | | | | |
| | R-1 | R-2 | R-3 | R-L | R-W | R-1 | R-2 | R-3 | R-L | R-W | R-1 | R-2 | R-3 | R-W | R-L |
| + AE ($C = 448$) | 0.337 | 0.123 | 0.059 | 0.297 | 0.112 | 0.337 | 0.115 | 0.051 | 0.289 | 0.108 | 0.283 | 0.113 | 0.057 | 0.270 | 0.097 |
| + AE ($C = 256$) | 0.323 | 0.109 | 0.048 | 0.285 | 0.105 | 0.323 | 0.101 | 0.041 | 0.277 | 0.101 | 0.272 | 0.103 | 0.048 | 0.262 | 0.093 |
| + AE ($C = 64$) | 0.250 | 0.048 | 0.011 | 0.226 | 0.077 | 0.234 | 0.047 | 0.011 | 0.227 | 0.077 | 0.195 | 0.047 | 0.013 | 0.199 | 0.064 |

## A.3 More Generated Summaries Examples

The samples of generated summaries using the weighted random sampling and beam search inference methods are presented in Tables 9, and 10. Both tables are using an autoencoder with a latent space size of 384 which showed promising results in our experiments. The generated summaries maximum length is set to 60 tokens in all experiments. All the generated summaries sound good and capture the main information of the original texts.

Table 9: Comparing the generated summaries of the vanilla model and the model with a latent space size of 384 using the weighted random sampling decoding method.

| Model | Vanilla Model Generated Summary (Weighted Random Sampling) | +AE Generated Summary (Weighted Random Sampling) |
| --- | --- | --- |
| Transformer | masked men armed with handguns have robbed three banks in pittsburgh area so far this year . they are believed to be armed and extremely dangerous . they are thought to have been armed with handguns and are thought to be from pittsburgh . the suspects are described as white , 5 ' 8 ' to 5 | the men , who are seen on surveillance footage pointing guns at bank employees ' heads , have threatened to kidnapping those at their targets and shoot police . however , the way that the two men handle their weapons has led the fbi to suspect that the thieves are actually former police officers themselves . they are also |
| BART | two men have been robbed by the fbi since april 10, according to surveillance footage. they have been seen holding his finger stretched along the barrel of his gun. they have been seen jumping over the counter as they begin their heists. the two robbers have a gun worn during the robberies | two pennsylvania bank robbery suspects have been seen in a string of recent heists. the suspects are believed to have been from pittsburgh. the suspects are believed to be from pittsburgh because of their attitudes. |
| BERT | the unknown men , who are seen on surveillance footage , have threatened to kidnap those at their targets and shoot police . the two men , both 5 ' 5 ' - 9 ' and april 10 , have also been taken to the bank in pittsburgh , pennsylvania . the fbi believes the two suspects may have | two pennsylvania bank robbers armed as they do a series of recent robberies . they have been described as ' armed and extremely dangerous ' . they have been seen on surveillance footage showing the two men . they have been described as ' armed and extremely dangerous ' and dangerous . |
| DistilBERT | the men , who wear dark sweatpants , are believed to be armed and extremely violent . the two men are thought to have been armed and armed . they are believed to be from pittsburgh , pennsylvania , who have been robbed three banks . the men are thought to have been wearing the gun and a gun . | two masked men are thought to have robbed three banks in pittsburgh this year . they are believed to have been armed and extremely dangerous . they have been described as armed and extremely dangerous . |

Table 10: Comparing the generated summaries of the vanilla model and the model with a latent space size 384 using the beam search decoding method.

| Model | Vanilla Model Generated Summary (Beam Search) | +AE Generated Summary (Beam Search) |
| --- | --- | --- |
| Transformer | masked men armed with handguns have robbed three banks in pittsburgh area so far this year , most recently on april 10 | two men armed with handguns robbed three banks in pittsburgh area so far this year , most recently on april 10 |
| BART | the men, who are seen on surveillance footage pointing their guns at bank employees' heads, have threatened to kidnapping those at their targets and shoot police. the two men are actually former police officers themselves. | two pennsylvania bank thieves are believed to have had military training and are being described by the fbi as 'armed and extremely dangerous'. |
| BERT | two pennsylvania bank robbers are believed to have had military training and are being described by | two pennsylvania bank robbers armed with handguns have been robbed in the pittsburgh area so far this year , most recently on april 10 |
| DistilBERT | a pair of masked men armed with handguns have robbed three banks in the pittsburgh area so far this year , most recently on april 10 . the unknown men , who are seen on surveillance footage pointing their guns at bank employees ' heads , have threatened to kidnapping and shoot police . | two masked men armed with handguns have robbed three banks in the pittsburgh area so far this year , most recently on april 10 |

## A.4 BERTSCORE RESULTS

The following table (Table 11) demonstrates the evaluation results of the proposed method using the BERTScore metric. The mentioned metric is also confirming the paper's discussions on contextual level.

Table 11: Comparing the vanilla and proposed models generated summaries quality using the BERTScore metric.

| Model | Inference Method | Vanilla | +AE ($C = 512$) | +AE ($C = 384$) | +AE ($C = 128$) | +AE ($C = 32$) |
|---|---|---|---|---|---|---|
| **Transformer** | *Greedy* | 0.858 | 0.861 | 0.860 | 0.846 | 0.801 |
| | *Random* | 0.857 | 0.860 | 0.859 | 0.847 | 0.809 |
| | *Beam* | 0.852 | 0.858 | 0.857 | 0.853 | 0.805 |
| **BART** | *Greedy* | 0.867 | 0.865 | 0.863 | 0.845 | 0.814 |
| | *Random* | 0.869 | 0.864 | 0.862 | 0.845 | 0.819 |
| | *Beam* | 0.866 | 0.866 | 0.865 | 0.851 | 0.821 |
| **BERT** | *Greedy* | 0.858 | 0.857 | 0.854 | 0.854 | 0.809 |
| | *Random* | 0.857 | 0.856 | 0.854 | 0.843 | 0.815 |
| | *Beam* | 0.841 | 0.854 | 0.854 | 0.846 | 0.814 |
| **DistilBERT** | *Greedy* | 0.798 | 0.855 | 0.855 | 0.842 | 0.809 |
| | *Random* | 0.836 | 0.855 | 0.854 | 0.844 | 0.815 |
| | *Beam* | 0.802 | 0.856 | 0.855 | 0.846 | 0.814 |

