# OpenReview forum: "Compressing Transformer-Based Sequence to Sequence Models With Pre-trained Autoencoders for Text Summarization"
_ICLR.cc/2022/Conference — ICLR 2022 Submitted_

### Official Review · Reviewer_1LU5 · 2021-10-28

**Correctness:** 3
**Technical Novelty And Significance:** 2
**Empirical Novelty And Significance:** 2
**Recommendation:** 3
**Confidence:** 4

**Main Review:**

This paper investigates the use of a pre-trained autoencoder to reduce the output dimensionality of a pre-trained transformer (such as BERT or BART) so that a lower dimensionality decoder can be used.

Three kinds of AE are investigated: Linear, LSTM and CNN.  The way these are used is not very clear but I assume that the linear AE maps each symbol independently whereas the LSTM and CNN see the whole sequence?  This point should be clarified.  Also, what motivated the choice of a purely linear network.  A standard MLP bottleneck autoencoder seems to me to be a more obvious choice than a CNN.  Why wasnt this used?


The AE is trained using "60% of these combinations of dataset" - please explain what motivates the choice of 60% and confirm that no test material is used for pre-training the autoencoder.

There appears to be a problem with Table 4 - perhaps the labelling in the left hand column?  In any event, I cant find the comparisons between AE, AE S and LL.  It's really not obvious why AE S should be worse than AE.  Please comment on this.

The results show that you can reduce the decoder size by up to 40% without sacrificing too much in performance.  However, this does nothing to change the size of the pre-trained encoder and it is the problem of the ever-increasing size of the pre-trained encoder which
is highlighted in the introduction to the paper.

It seems to me that what you are really doing here is adjusting the size of the decoder to best match the amount of training data for the downstream task.  In your case, text summarisation.  A different task with much more training data might give a different result.  Please comment on this.

**Summary Of The Paper:**

This paper investigates the use of a pre-trained autoencoder to reduce the output dimensionality of a pre-trained transformer (such as BERT or BART) so that a lower dimensionality decoder can be used.  The results show that for a summarisation task, the decoder size can be reduced by 40% without significant loss of accuracy as measured by rouge.

**Summary Of The Review:**

This paper describes a simple approach to reducing the dimensionality of the decoder to be used for a pre-trained encoder.  The experimental results may be of interest to practitioners concerned with reducing model sizes.  However, the original contribution is rather small and the restriction to a single downstream task leaves questions over its general applicability.

---

> ### Author Response · Authors · 2021-11-17
> **Response to Reviewer 1LU5**
>
> Thank you for reviewing our paper and pointing out possible improvement. It seems that there are some misunderstandings due to lack of clear explanation in the paper. We hope that the below description and the changes to the submission can clear out some confusions.
>
> ---
>
>
> * **Q** The way these are used is not very clear but I assume that the linear AE maps each symbol independently whereas the LSTM and CNN see the whole sequence? This point should be clarified. Also, what motivated the choice of a purely linear network. A standard MLP bottleneck autoencoder seems to me to be a more obvious choice than a CNN. Why wasnt this used?
>
>   * **4.1**   The autoencoder models are trained to compress the embedding size of different pre-trained encoders (for example; BERT) from [batch_size, seq_len, 768] to [batch_size, seq_len, compression_size]. (Added some explanation to the paper as well) Also, our linear autoencoder is not a single linear layer network. It is an MLP with 6 layers. (as stated in section 3.2 and figure 3)
>
>
> * **Q** please explain what motivates the choice of 60% and confirm that no test material is used for pre-training the autoencoder.
>
>   * **4.2**  The idea is to hold back on some training data for the summarizer model to ensure that our proposed method can also compress the encoder’s representation on unseen data. Also, the test set is selected from the CNN/DM dataset and it has not been used while training neither the AE nor the summarizer model.
>
>
> * **Q** There appears to be a problem with Table 4 - perhaps the labelling in the left hand column? In any event, I cant find the comparisons between AE, AE S and LL. It's really not obvious why AE S should be worse than AE. Please comment on this.
>
>   * **4.3**  Unfortunately, the labels were not correct in Table 4. We fixed the issue and appreciate that you point out the mistake.
>
>     The AE models are trained independently to compress the data to a smaller latent space and reconstruct it using the said representation. They learn how to keep the most useful information from the mentioned data using an MSE loss. However, the AE S model is basically a 3-layer MLP on top of the summarizer model’s encoder (for example, BERT) that tries to project the data to a smaller space and uses the summarizer model’s loss function (CrossEntropy). So, it appears to be difficult to train the summarizer model and the AE S jointly.
>
>
> * **Q** The results show that you can reduce the decoder size by up to 40% without sacrificing too much in performance. However, this does nothing to change the size of the pre-trained encoder and it is the problem of the ever-increasing size of the pre-trained encoder which is highlighted in the introduction to the paper.
>
>   * **4.4** We are focusing on the text summarization task where the number of parameters in both encoder and decoder will add up. In this experiment, we solely focused on the decoder component, and it was not our intention to reduce the encoder size.
>
>
> * **Q** It seems to me that what you are really doing here is adjusting the size of the decoder to best match the amount of training data for the downstream task. In your case, text summarization. A different task with much more training data might give a different result. Please comment on this.
>
>   * **4.5**   For training the autoencoder, we can have access to a lot of data because the training is unsupervised. Once the autoencoder model (that can do the dimensionality reduction) is trained, the amount of supervised data for fine-tuning the model could be increased/decreased as desired by the users. Of course, in general, more data will always help get better results in deep learning, but it will not have a negative effect.

---

> > ### Comment · Reviewer_1LU5 · 2021-11-23
> > **Unchanged review**
> >
> > I thank the authors for their responses to my questions which have clarified some of the confusions.  However, I remain of the opinion that there is insufficient novelty and practical significance in this paper to warrant acceptance.

---

### Official Review · Reviewer_wmem · 2021-10-29

**Correctness:** 2
**Technical Novelty And Significance:** 2
**Empirical Novelty And Significance:** 2
**Recommendation:** 3
**Confidence:** 4

**Main Review:**

Strengths

1. The paper was written clearly enough to understand the basic ideas.
2. The paper conducted extensive experiments to find the ideal trade-off between the compression ratio and model’s text generation capability.

Weaknesses:

1. The results of the transformer baseline is far from previous literatures, such as [Li et al., 2018; Lewis et al., 2019; Wei et al., 2020] that achieves at least 33.42 R-L score, while this paper report a 31.2 R-L score with the transformer baseline.

2. There is no any comparisons with existing compression methods, including pruning and distillation.

[Li et al., 2018] Improving Neural Abstractive Document Summarization with Explicit Information Selection Modeling.
[Lewis et al., 2019] BART: Denoising Sequence-to-Sequence Pre-training for Natural Language Generation, Translation, and Comprehension.
[Wei et al., 2020] Multiscale Collaborative Deep Models for Neural Machine Translation.

**Summary Of The Paper:**

This paper proposes to compress the transformer-based summarization models with well pre-trained autoencoders (AE). In this architecture, AE produces an intermediate states that is obtained by compressing the encoder’s final representation and pass it to the decoder.

**Summary Of The Review:**

 In this paper, the baseline system is far from literatures and is lack of comparisons to existing methods.

---

> ### Author Response · Authors · 2021-11-17
> **Response to Reviewer wmem**
>
> Thank you for reviewing our paper and pointing out possible improvement. It seems that there are some misunderstandings due to lack of clear explanation in the paper. We hope that the below description and the changes to the submission can clear out some confusions.
>
> ---
>
>
> * **Q** The results of the transformer baseline is far from previous literatures, such as [Li et al., 2018; Lewis et al., 2019; Wei et al., 2020] that achieves at least 33.42 R-L score, while this paper report a 31.2 R-L score with the transformer baseline.
>
>   * **3.1**  Our transformer model is a small vanilla transformer model with 6 encoder/3 decoder layers. It has only 70M parameters which is significantly smaller than other models. It is just a baseline to evaluate the proposed method’s effectiveness in comparison to the effectiveness of a custom small model and is not designed to reproduce any specific paper’s results. We add more explanation about this on the paper to reduce the confusion.
>
>
> * **Q** There is no any comparisons with existing compression methods, including pruning and distillation.
>
>   * **3.2**  Please refer to Answer 1.3 with [response to reviewer Lsjg](https://openreview.net/forum?id=QevkqHTK3DJ&noteId=TGTFlOq_SDG).

---

### Official Review · Reviewer_vaDv · 2021-11-02

**Correctness:** 3
**Technical Novelty And Significance:** 2
**Empirical Novelty And Significance:** 2
**Recommendation:** 5
**Confidence:** 3

**Main Review:**

Strength
1. The paper is clearly written. The paper tries to discover the trade-off between the compression ratio and model performance. The paper shows experiment results by comparing different types of autoencoders as well as different compression rates. The paper also presents the generation results and another related experiment in the appendix.

Weakness
1. How is the compression rate determined? It would be better to show why 32, 128, 384,512 are chosen for the decoder size. The scale of those dimensions is not even linear. It might be more reasonable for readers to see dimensions with 32, 64, 128, 256, 512. The paper needs to include more comparison between other compression methods such as distillation, information bottleneck, pruning. The paper fails to show insight qualitative analysis for the Table. 3 and Table 4. Table 4 seems to be incomplete. In Section 3.4, why not use all training data to train the autoencoder.

2. The rouge scores might not be good  It would be better to incorporate other metrics such as BLEU, BERTscore.

**Summary Of The Paper:**

The paper proposes a new autoencoder-based seq2seq model for text summarization tasks. The paper tries to find the best trade-off between compression ratio and model performance. The paper conducts extensive experiments by evaluating the loss of accuracy with ROUGE.

**Summary Of The Review:**

Overall, this paper presents an interesting experiment to discover the trade-off between the compression rate and the performance. The experiment seems a little bit naive without detailed analysis.  I recommend rejection for this paper.

---

> ### Author Response · Authors · 2021-11-17
> **Response to Reviewer vaDv**
>
> Thank you for reviewing our paper and pointing out possible improvement. It seems that there are some misunderstandings due to lack of clear explanation in the paper. We hope that the below description and the changes to the submission can clear out some confusions.
>
> ---
>
> * **Q** How is the compression rate determined? It would be better to show why 32, 128, 384,512 are chosen for the decoder size. The scale of those dimensions is not even linear. It might be more reasonable for readers to see dimensions with 32, 64, 128, 256, 512.
>
>   * **2.1**  We did more experiments on the BERT model with three more compression rates. (Appendix A.2) The complete list of compression sizes is 32, 64, 128, 256, 384, 448, and 512. Our results showed the same trend in all experiments.
>
>
> * **Q** The paper needs to include more comparison between other compression methods such as distillation, information bottleneck, pruning.
>
>   * **2.2**  Please refer to Answer 1.3 with [response to reviewer Lsjg](https://openreview.net/forum?id=QevkqHTK3DJ&noteId=TGTFlOq_SDG).
>
>
> * **Q** The paper fails to show insight qualitative analysis for the Table. 3 and Table 4. Table 4 seems to be incomplete.
>
>   * **2.3**  Unfortunately, the labels were not correct in Table 4. We fixed the issue and appreciate that you point out the mistake.
>
>
> * **Q** In Section 3.4, why not use all training data to train the autoencoder.
>
>   * **2.4**  The idea is to hold back on some training data for the summarizer model to ensure that our proposed method can also compress the encoder’s representation on unseen data. We tried to make it more clear in the paper on the latest revision.
>
>
> * **Q** The rouge scores might not be good It would be better to incorporate other metrics such as BLEU, BERTscore.
>
>   * **2.5**  While the ROUGE score is the dominant metric for evaluating text summarization tasks, we added the BERTScore metric evaluation to the paper as well. Thanks for the suggestion.

---

### Official Review · Reviewer_Lsjg · 2021-11-03

**Correctness:** 2
**Technical Novelty And Significance:** 2
**Empirical Novelty And Significance:** 2
**Recommendation:** 3
**Confidence:** 3

**Main Review:**

Weaknesses
1. The motivation for using autoencoder is not quite clear. Actually, BART is already pretrained with a denoising loss which should be better than autoencoder loss. How about a more detailed comparison between these two losses?
2. It's not clear whether the encoder is frozen or not. Is the encoder further optimized when fine-tuning? If not, it's not clear why not optimize encoder. If optimized, then pre-training losses from BART can also be adopted.
3. Missing some knowledge distillation baselines, such as Noisy Self-Knowledge Distillation for Text Summarization and PRE-TRAINED SUMMARIZATION DISTILLATION
4. The paper writing can be further improved. Figures 1-3 have been shown in many papers. It would be better to emphasize more on the novel part.


**Summary Of The Paper:**

The authors propose to add an autoencoder on top of a pre-trained encoder to reduce the encoder’s output dimension and allow to significantly reduce the size of the decoder.

**Summary Of The Review:**

Overall, the proposed loss is not novel enough and needs to have a further comparison with other pertaining losses such as the losses from BART. And also missing some strong Seq2Seq baselines with knowledge distillation losses.

---

> ### Author Response · Authors · 2021-11-17
> **Response to Reviewer Lsjg**
>
> Thank you for reviewing our paper and pointing out possible improvement. It seems that there are some misunderstandings due to lack of clear explanation in the paper. We hope that the below description and the changes to the submission can clear out some confusions.
>
> ---
>
> * **Q** The motivation for using autoencoder is not quite clear. Actually, BART is already pre-trained with a denoising loss which should be better than autoencoder loss. How about a more detailed comparison between these two losses?
>
>   * **1.1** The Autoencoder has been used as a dimensionality reduction method, where its encoder projects the input to a smaller latent space, and the decoder part tries to recreate the input. It is responsible for reducing a pre-trained encoder’s (e.g. BERT) output dimension, resulting in a smaller decoder size in our summarizer model. The reasoning for this size reduction is that the summarizer model decoder will have to process smaller tensors.
>
>
> * **Q** It's not clear whether the encoder is frozen or not. Is the encoder further optimized when fine-tuning? If not, it's not clear why not optimize the encoder. If optimized, then pre-training losses from BART can also be adopted.
>
>   * **1.2**  We have two different encoders in this experiment:
>     1.	The Auntoencoder’s encoder ($ AE_{enc} $): Takes care of dimensionality reduction. We train an autoencoder for each pre-trained transformer model (for example, BERT), then freeze the Autoencoder’s encoder and use it in our summarizer model. The reason for freezing it is to reduce the number of trainable parameters. Our experiment showed that training it jointly with the summarizer model will reduce the ROUGE score. (AE S)
>     2.	Transformer encoder ($ T_{enc} $): It is chosen from a number of well-known pre-trained models. It is also frozen in our summarizer model. We chose to do it to eliminate an additional fine-tuning factor from the experiments. It could have raised questions like if different models based on their sizes should be treated differently. The focus of this paper is to show that a Linear Autoencoder can reduce a pre-trained encoder representation size while not losing critical information for the decoder to generate a meaningful summary.
>
>     A number of changes have been done to the paper and figures to better clarify this for readers.
>
>
> * **Q** Missing some knowledge distillation baselines, such as Noisy Self-Knowledge Distillation for Text Summarization and PRE-TRAINED SUMMARIZATION DISTILLATION
>
>   * **1.3**  While there are numerous techniques available to reduce the network size, we believe these methods are orthogonal to each other and can be used together. We used the combination of knowledge distillation (DistilBERT) and our proposed AE method to reduce the network size even more and increase the R-1 score by 5%. The same concept can be applied to other distilled models.
>
>
> * **Q** Figures 1-3 have been shown in many papers. It would be better to emphasize more on the novel part.
>
>   * **1.4**  We rearranged the figures in the paper to better describe our approach and removed the unnecessary ones.

---

### Decision · Program_Chairs · 2022-01-20

**Decision:**

Reject

**Comment:**

The paper proposes to incorporate an autoencoder to transformer-based summarization models in order to compress the model while preserving the quality of summarization. The strengths of the paper, as identified by reviewers, are in extensive experiments presented in the paper and in a relatively clear write-up. However, the reviewers identify several weaknesses, including missing state-of-the-art summarization baselines and missing relevant compression/knowledge distillation baselines. Although the author response have addressed some of reviewers' concerns, all the reviewers agree that the draft is not yet ready for publication.